# Phytochemical Substances—Mediated Synthesis of Zinc Oxide Nanoparticles (ZnO NPS)

Fawzeeh Nayif Alharbi [1], Zulfa Mohamed Abaker [2,3,*] and Suzan Zein Alabdeen Makawi [2,4]

1  Department of Chemistry, College of Science, Qassim University, Buraidah 51452, Saudi Arabia; 411207262@qu.edu.sa
2  Department of Chemistry, College of Science and Arts, Qassium University, Ar Rass 51921, Saudi Arabia; s.alkhaleefa@qu.edu.sa
3  Department of Chemistry & Industrial Chemistry, College of Applied and Industrial Sciences, Bahri University, P.O. Box 12327, Khartoum 11111, Sudan
4  Department of Chemistry, Faculty of Science and Technology, Alneelain University, Khartoum 11121, Sudan
*  Correspondence: zulfa.abaker@gmail.com

**Abstract:** *Artemisia absinthium* (*A. absinthium*) leaf extract was successfully used to create zinc oxide nanoparticles (ZnO NPs), and their properties were investigated via several techniques, including X-ray diffraction (XRD), scanning electron microscopy (SEM), energy-dispersive X-ray (EDX), Fourier transform infrared (FTIR), and ultraviolet–visible spectroscopy (UV–Vis spectroscopy). SEM analysis confirmed the spherical and elliptical shapes of the particles. Three different zinc peaks were observed via EDX at the energies of 1, 8.7, and 9.8 keV, together with a single oxygen peak at 0.5 keV. The XRD analysis identified ZnO NPs as having a hexagonal wurtzite structure, with a particle size that decreased from 24.39 to 18.77 nm, and with an increasing surface area (BET) from 4.003 to 6.032 $m^2/g$ for the ZnO (without extract) and green ZnO NPs, respectively. The FTIR analysis confirmed the groups of molecules that were accountable for the stabilization and minimization of the ZnO NPs, which were apparent at 3400 cm. Using UV–Vis spectroscopy, the band-gap energies (Egs) for the green ZnO and ZnO (without extract) NPs were estimated, and the values were 2.65 and 2.79 eV, respectively.

**Keywords:** *Artemisia absinthium*; green ZnO NPs; band-gap energy; plant extract; phytochemicals

## 1. Introduction

The most recent fields to develop and expand rapidly are nanoscience and nanotechnology. Nanomaterials are used in a variety of industries, including in the electrical and electronics, textile, cosmetic, and medicinal sectors. Nanomaterials are materials that are generated using nanotechnologies; these include nanoparticles (NPs) with sizes between 1 and 100 nm. In industrial settings, metal NPs and metal oxides are commonly required. Some of the different types of metal and metal oxide NPs with multiple uses are aluminum, nickel, silver, copper, copper oxide, iron, iron oxide, cerium dioxide, titanium dioxide, and zinc oxide [1–3]. NPs are produced using a number of physical, chemical, and biological methods; however, these chemical and physical processes frequently require a lot of energy, and can produce dangerous and poisonous substances that can result in additional risks [4]. In order to deal with these issues, current researchers have created biotechnology or "green" technology, which uses plant substances with low levels of the chemicals as a reliable, inexpensive, and safe synthesis method.

ZnO NPs have attracted more interest compared to the other metal oxides, due to their safe and low-cost manufacturing and preparation methods [5,6]. ZnO has numerous uses in the domains of engineering, biology, and medicine. Many engineering applications exist for ZnO NPs, including solar cells applications; due to their long-term durability, strong conductivity, high electron affinity, and excellent electron mobility [7]; and photodetectors,

biosensors [8], chemical sensors [9], and gas sensors. Furthermore, ZnO NPs exhibit cytotoxic, antibacterial, and fungicidal properties in biological and medical applications [10,11]. They also have chemiluminescent properties [12], and show wound-healing, antidiabetic and antiinflammatory activities [10,13]. The most important applications of ZnO NPs are mostly in wastewater treatment, which needs to be conducted with a reuse application, to meet the increasing need for water in both the agricultural and industrial sectors. Because of their outstanding durability and aromatic nature, physical and biological treatments typically do not successfully remove dyes from wastewater. A photocatalysis degradation method, which is able to eliminate contaminants, particularly dyes, from wastewater, is one of the promising, easy-to-use, and cost-effective solutions. ZnO, among others metal oxides, exhibits a high potential in pollution remediation, due to its significant photocatalytic activity under sunlight [14]. ZnO, a material that is capable of displaying a variety of nanostructures, has exceptional semiconducting, visual, and dielectric capabilities. As a result, research has been conducted on ZnO-based nanomaterials for a variety of uses, including electronic and optical devices, energy storage, cosmetics, nanosensors, etc. [15]. ZnO has a broad band-gap semiconductance (3.37 eV), and a high excited-state binding energy (60 meV), which lead to its extremely efficient excitonic blue and near-UV emission [15,16]. ZnO has been given Food and Drug Administration (FDA) approval for use in sunscreens, due to its stability and innate ability to absorb ultraviolet (UV) radiation [17,18].

The method of choice for nanoparticle synthesis is plant-based, which is simple to generate and establish [19]. The synthesis of NPs, particularly using phytochemicals, is a recent development that is considered straightforward, affordable, and harmless [20]. Conventional methods for the production of nanoparticles have disadvantages; namely, their lengthy processing times, costs, and usage of hazardous substances. Due to these restrictions, the majority of relevant studies have concentrated on green and quick synthesis techniques for the creation of nanoparticles [21,22]. Plant-based techniques have been accepted to be environmentally friendly, as these techniques create items and byproducts that are eco-friendly. Additionally, they use less energy, do not utilize expensive chemicals, and produce more; thus, green methods are promoted as being economical [23]. Due to the availability of phytochemicals and numerous bioactive compounds with numerous functional groups, including polyphenols, flavonoids, terpenoids, carboxylic acids, quinones, aldehydes, ketones, and amides, phytochemicals are capable of being utilized to create nanoscale reduction agents [24,25]. These phytochemicals reduce metal ions into nano form, through a reduction mechanism [26]. Metal oxide nanoparticle stability, controllability of size and shape, and prevention of aggregation are all strongly affected by phytochemicals. It is widely recognized that phytochemicals serve as reducing and capping agents during the green production of nanoparticles [27]. When creating nanoparticles, capping agents play an essential role as a dependable stabilizer to prevent excessive particle development. These substances maintain the stability of nanoparticle surfaces, to prevent species interactions in the synthesis medium. The stabilizing agents can affect the biological characteristics of the nanoparticles [28]. In order to communicate successfully with the target cells for therapeutic purposes, they can modify the surface properties of nanoparticles. In these situations, the capping agents need to be non-toxic, and biocompatible with living organisms for consumption [29].

Many types of plants have been used in the manufacturing of ZnO NPs; namely, aloe vera [30], moringa oleifera leaf extract [31], Ocimum basilicum [32], rosemary leaves [33], azadirachta indica leaves [34], Lycopersicon esculentum (tomato) [35,36], etc.

*A. absinthium* is among the therapeutic plants that are used in traditional healthcare, and numerous studies have been conducted on it, to determine the extent to which it has inhibitory and antibacterial properties. A thorough review of the literature on *A. absinthium*'s phytochemical data reveals that their principal constituents are polyphenolics, terpenoids, flavonoids, coumarins, caffeoylquinic acids, sterols, and acetylenes, which are responsible for the reduction process [37–39].

The main objective of this investigation was to produce ZnO nanostructures, utilizing the fresh leaves of *A. absinthium* extract as a dispersion and reduction agent. Additionally, *A. absinthium* extract and ultrasonic energy were combined, to check for any product alterations. The nanoparticles' presence was then verified, using XRD, SEM, EDX, FTIR, and UV–Vis. This study purposely focused on ZnO NPs, as ZnO is one of the most in-depth-researched semiconductor-type metal oxides. The significance of this research lies in the potential applications of the greenly manufactured ZnO nanoparticles from *A. absinthium* plant leaf extract, in the treatment of water pollution (dye removal); photocatalytic activity; pharmaceutical (antibacterial) applications; cosmetic, anticancer, and antidiabetic activity; and use in the opto-electronic (solar cell), concrete, and rubber industries.

## 2. Materials and Methods

### 2.1. Materials

*A. absinthium* leaves were collected from a home garden in Al-Qassim Al-Rass, Saudi Arabia. Zinc nitrate hexahydrate (Zn $(NO_3)_2 \cdot 6H_2O$) was obtained from LOBA Chemie, and potassium hydroxide (KOH) was obtained from Techno Pharmchem, India.

### 2.2. Methods

#### 2.2.1. Plant Extract

The method was adapted from Rasli et al. [30] and M. Ali et al. [40], with some modifications. Approximately 10 g of fresh *A. absinthium* leaves was cleaned with distilled water, before being chopped into pieces and crushed, with 100 mL of distilled water, into a slurry, with the use of a pastille. The mixture was heated at 70 °C for 30 min, with a magnetic stirrer being used. The extract was allowed to cool to ambient temperature, before being filtered through Whatman No. 1 filter paper, and kept in a refrigerator at 4 °C for use in the subsequent experiments (regarding reducing and capping agents).

#### 2.2.2. Green Synthesis of ZnO NPs

As show in Figure 1., ZnO NPs were created using a direct precipitation technique. Aqueous solutions were prepared using Zn $(NO_3)_2 \cdot 6H_2O$ (10 g in 100 mL of deionized water) and 0.5 M KOH (2.8 g in 100 mL of deionized water). A small amount of KOH was gradually added to 90 mL of Zn $(NO_3)_2 \cdot 6H_2O$, to bring the mixture up to pH 12. A volume of 10 mL of plant aqueous extract was added, while being vigorously stirred, and kept at a temperature of 90 °C, using a magnetic stirrer, for 2 h, until the suspension was created. After that, it was sonicated at 500 Hz for 20 min, resulting in a yellow precipitate. The precipitate was repeatedly rinsed with water and ethanol before being dried in a hot-air oven for an entire night at 90 °C. The precipitate was calcined for 2 h at 500 °C in a muffle furnace. The same method was repeated with Zn $(NO_3)_2 \cdot 6H_2O$, without adding plant extract. The method is adapted from M. Ali et al., with a few modifications [40].

### 2.3. Characterization Techniques

#### 2.3.1. Characterization of Artemisia absinthium Extract

Two techniques were used to detect the active components in the plant extract: reagent-based qualitative analysis, followed by drying, and analysis with FTIR (SHIMADZU).

#### Flavonoids Test

A volume of 1 mL of sodium hydroxide (NaOH) was added to 1 mL of plant extract; the development of a bright yellow color provides evidence of the existence of flavonoids.

#### Wagner's Test

A volume of 1 mL of 1.5% *v/v* hydrochloric acid (HCl) was used to acidify the plant extract, before Wagner's reagent was added. The production of a brown color served as a sign that alkaloids were present.

Frothing Test

A volume of 1 mL of the plant extract was diluted individually with 5 mL of distilled water, and shaken for 15 min. The formation of a thick layer of foam shows evidence of saponins.

Ferric Chloride Test

Neutral ferric chloride (FeCl₃), 5%, was added to 1 mL of the plant extract. The presence of tannins and phenolic compounds is indicated by the production of a dark blue or bluish-black colored product.

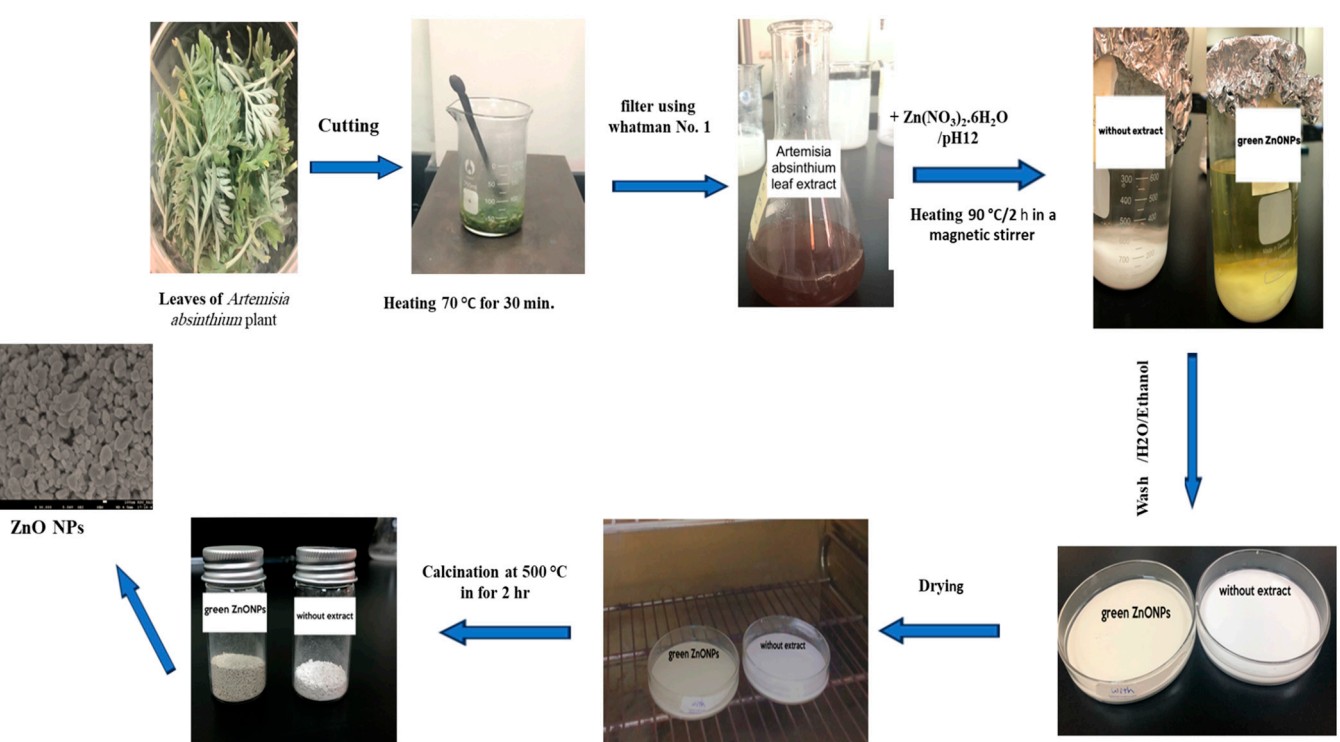

**Figure 1.** Green synthesis of ZnO NPs.

2.3.2. Characterization of ZnO NPs

In order to investigate the ZnO NPs, several analytical methods were adopted.

XRD—Diffraction Analysis

An X-ray diffractometer (XRD, Rigaku with K beta filter, time duration 10.000°/ min., scanning range 10.0–90.0°, and operated at 40 kV, 40 mA) was used to examine the crystal size. The well-known Scherer formula was used to calculate the typical crystal size, D:

$$D = \frac{K\lambda}{\beta \cos \theta} \tag{1}$$

Here, λ is the wavelength (0.154 nm); β is the full width at half-maximum (FWHM) in radians, and K is a constant equal to 0.90 [41]; θ is the diffraction angle.

SEM and EDX Analysis

EDX analysis was used to determine the elementary calculation, while SEM (FESEM, JEOL-SEM, 6700F) was used to examine the surface morphology of the NPs.

FTIR Analysis

A Perkins Elmer FTIR spectrometer (4000–400 cm$^{-1}$) was used to identify the functional groups, using the KBr technique.

UV–Vis Analysis

A UV-2550 (Shimadzu, Tokyo, Japan), with a scanning range from 200 to 800 nm, was used to monitor the diffuse reflection/absorption spectra (DRS), and to calculate the band-gap energy.

## 3. Results and Discussion

### 3.1. Characterization of A. absinthium Leaf Extract

3.1.1. Identification of Active Ingredients

The generated plant extract works as a stabilizing, as well as a decreasing, agent; it contains a large amount of polyphenols, which, in turn, consist of flavonoids, antibiotics, antioxidants, and organic aggregates. When this extract is added to zinc salt, it breaks the hydroxyl (OH) bond, and forms a partial bond with the metal; when this partial bond is broken, the electrons move to form zinc hydroxide, which in turn reacts with (OH) coming from sodium hydroxide, to form nanoscale zinc oxide. Due to the availability of the OH groups for the production of NPs, flavonoids and tannins are the primary phytochemical component of *Artemisia absinthium* extract, which are visible bioactive minimizing and stabilizing agents [42]. The *Artemisia absinthium* extract was subjected to phytochemical analysis, using a variety of reagents to identify several types of bioactive substances. The most significant active components found are outlined in Table 1. Figure 2. explains the role of the plant extract in reducing a metal salt into nanoparticles.

**Table 1.** Phytochemical analysis of the plant extract.

| No. | Active Components | Test | Result |
|:---:|:---:|:---:|:---:|
| 1 | Phenolics | FeCl$_3$ | + |
| 2 | Alkaloids | Wagner's reagent | + |
| 3 | Saponins | foam test | + |
| 4 | Flavonoids | alkaline test | + |

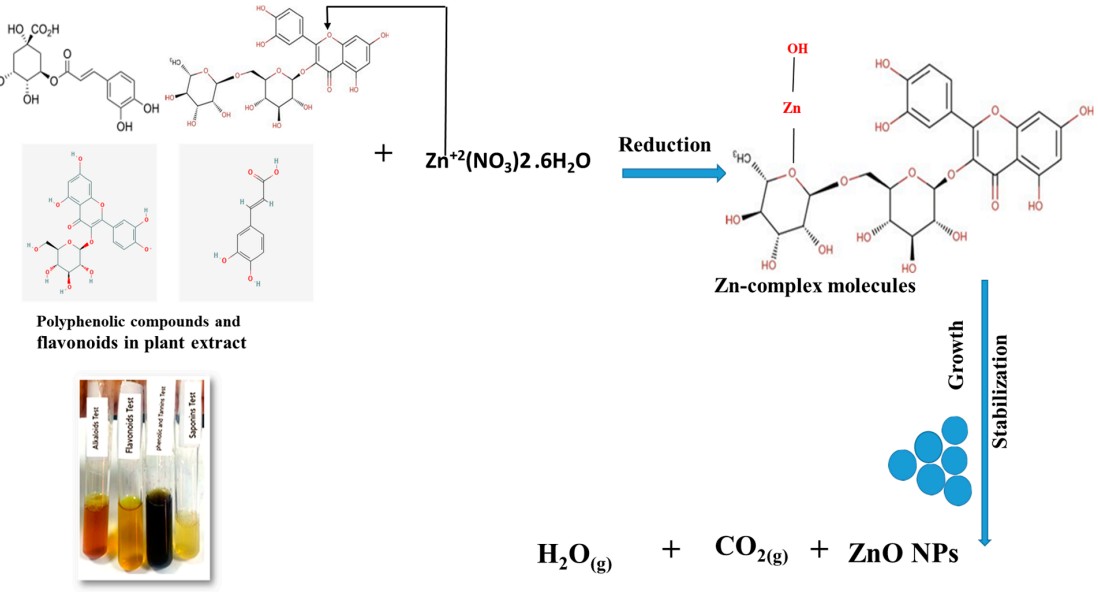

**Figure 2.** Reduction of the active components to zinc salts.

### 3.1.2. FTIR Analysis

The spectrum of the plant extract was determined after it was dried from water at room temperature. Figure 3 represents the infrared peaks of the plant extract. The results show that the absorption bands at 3400, 1608, and 1063 cm$^{-1}$ belong to the stretching vibrations of OH, C=O, and C-O, respectively, which confirmed the presence of polyphenolic substances that can serve as minimizing and stabilizing agents.

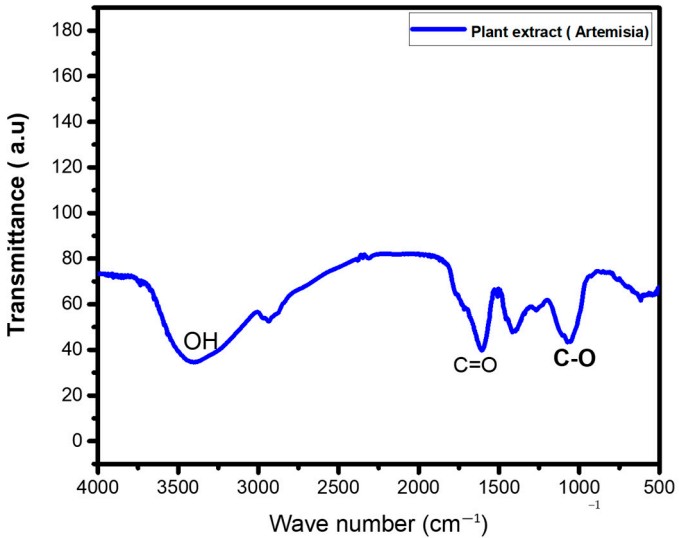

**Figure 3.** FTIR analysis of the plant extract.

### *3.2. Characterization of ZnO NPs*

### 3.2.1. X-ray Diffraction

The materials' particle size and crystallinity were evaluated using XRD analysis. The structural properties of the prepared NPs are shown in Figure 4. Compared with the data from JCPDS Card No. 03-065-0725, there was no sign observed of a peak impurity or secondary phase, and a hexagonal (wurtzite) ZnO structure has been observed. The strong and narrow diffraction peaks, especially (100), (002), and (101), showed a good crystal structure, and high-quality peak intensities. Sharp extreme peaks of about 2θ at the numbers 31.84, 34.49, 36.32, 47.62, 56.68, 62.95, 66.48, 68.04, and 69.17 corresponded to the planes of (100), (002), (101), (102), (110), (103), (200), (112), and (201) orientations, respectively, for the ZnO NPs (without extract); meanwhile, the 2θ values for green ZnO appeared at 31.82°, 34.47°, 36.30°, 47.60°, 56.67°, 62.93°, 68.03°, and 69.154°, corresponding to the planes of the (100), (002), (101), (102), (110), (103), (200), (112), and (201) orientations, respectively.

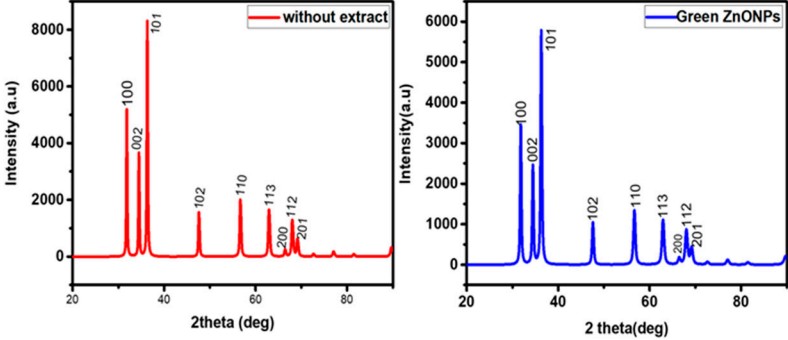

**Figure 4.** XRD spectra of the without-extract and green ZnO NPs.

The strong peak in direction (101) indicated that the nanomaterial prepared was in the hexagonal wurtzite phase. These findings demonstrated similar types of peak indices for the crystalline nature of ZnO NPs as were produced in the study carried out by [43].

The average crystallite sizes of the ZnO NPs (without extract) were calculated from the XRD data in Figure 4, using the Scherrer formula. The calculated crystallite size values are listed in Table 2.

**Table 2.** Particle sizes of the ZnO NPs (without extract) from Figure 4.

| No. of Peaks | Indices | Location (2θ) | FWHM (2θ) | Size (nm) |
|---|---|---|---|---|
| 1 | 100 | 31.8405 | 0.25611 | 32.25 |
| 2 | 002 | 34.4993 | 0.27379 | 30.38 |
| 3 | 101 | 36.3291 | 0.28256 | 29.59 |
| 4 | 102 | 47.6240 | 0.34103 | 25.46 |
| 5 | 110 | 56.6831 | 0.3981 | 22.67 |
| 6 | 113 | 62.9548 | 0.43834 | 21.25 |
| 7 | 200 | 66.4807 | 0.48187 | 19.71 |
| 8 | 112 | 68.0481 | 0.48285 | 19.85 |
| 9 | 201 | 69.1720 | 0.52439 | 18.40 |
| | | | Average size | 24.39 |

The average crystallite sizes of the green ZnO NPs were calculated from the XRD data in Figure 4, using the Scherrer formula. The calculated values for crystallite size are listed in Table 3.

**Table 3.** Particle sizes of the green ZnO NPs from Figure 4.

| No. of Peaks | Indices | Location (2θ) | FWHM (2θ) | Size (nm) |
|---|---|---|---|---|
| 1 | 100 | 31.8216 | 0.34989 | 23.61 |
| 2 | 002 | 34.47333 | 0.37528 | 22.16 |
| 3 | 101 | 36.30893 | 0.37771 | 22.13 |
| 4 | 102 | 47.60253 | 0.4578 | 18.96 |
| 5 | 110 | 56.67089 | 0.54533 | 16.55 |
| 6 | 113 | 62.93347 | 0.60605 | 15.36 |
| 7 | 200 | 66.56221 | 0.66134 | 14.37 |
| 8 | 112 | 68.03501 | 0.69904 | 13.71 |
| 9 | 201 | 69.15222 | 0.43528 | 22.16 |
| | | | Average size | 18.77 |

The calculated average crystallite size values of ZnO NPs (without extract) and green ZnO NPs were 24.39 and 18.77, respectively. It was observed that the average crystallite size was reduced in the green ZnO NPs. The reason for the reduction in size is the influence of phytochemical compounds that control, develop, and stabilize the crystals, as well as their slower rate of production, which has led scientists to develop synthesis techniques that allow for more accuracy in controlling the size and shape in a variety of applications. Additionally, green ZnO NPs have been found to have a small boost in surface area in relation to a decrease in size, which results in an increase in the photocatalytic activity, and the study's findings are consistent with those of earlier research [40].

The lattice parameters are the quantities specifying a unit cell or the unit of the periodicity of the atomic arrangement. The lattice parameters (constants) are composed of "a, b, c" lengths of the unit cell in three dimensions, and "$\alpha$, $\beta$, $\gamma$", their mutual angles. They are considered to comprise one of the most significant structural features that could influence physical properties, and can be studied using X-ray diffraction (XRD) technique [44].

To obtain the lattice parameters for the hexagonal (wurtzite) ZnO structure, the following equations are used [45].

$$a = \frac{\lambda}{\sqrt{3}sin\theta100} \qquad (2)$$

$$c = \frac{\lambda}{sin\theta002} \qquad (3)$$

The lattice parameters were calculated, and the values are shown in Table 4. A small increase may not have caused the XRD peaks to shift in position.

**Table 4.** Lattice parameters values of ZnO (without extract) and green ZnO.

| Sample | 2θ (100) | 2θ (002) | Lattice Parameters | |
|---|---|---|---|---|
| | | | a = b | c |
| ZnO (without extract) | 31.8405 | 34.4993 | 3.2439 | 5.1959 |
| Green ZnO | 31.8216 | 34.4733 | 3.2451 | 5.1994 |

### 3.2.2. FTIR Analysis

Figure 5 shows the resulting nanoparticles' spectra, analyzed using FTIR. The spectrum green ZnO NPs showed a significant decrease in the intensity of the peak at around 3400 cm$^{-1}$, compared with the spectrum of the plant extract (Figure 3); this indicates the vital role of the biomolecules attributed to this functional group in minimizing ZnO. The new broad absorption bands observed at 439.77 and 432.05 cm$^{-1}$ for the ZnO (without extract) and green ZnO NPs, respectively, support ZnO NP production. These findings are consistent with those of earlier research [46]. The peaks at 1388.74 and 1375.42 cm$^{-1}$, however, belong to the OH bending from phenol or tertiary alcohol. The peaks at 2333.86 and 2339.65 cm$^{-1}$ belong to the COO$^-$ group; to clarify the identity of this group, the molecule's subsequent carboxylic acids (COOHs) decreased, lost an atom of hydrogen, and created carboxylate ions (COO-). To enable them to serve as a surfactant, and stabilize the metal nanoparticles through electro-steric stabilization, the COO- ions and the remaining sequent were attached to the surface of the metal nanoparticles [5].

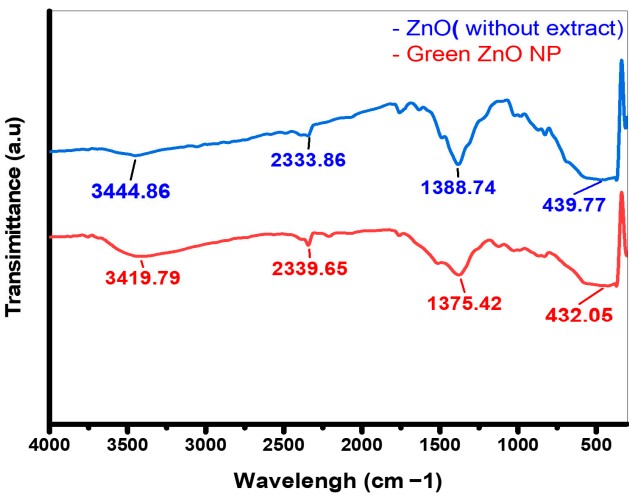

**Figure 5.** FTIR spectra of the without-extract ZnO and green ZnO NPs.

### 3.2.3. SEM Analysis

The spectra that resulted from the examination of the surface morphology (shape) are shown in Figure 6. The observed outcomes made it abundantly evident that the ZnO without extract and the green ZnO NPs showed different agglomerated particles. The low-magnification pictures captured in Figure 6a,c clearly show that the aggregated particles did not entirely separate, whereas those captured at higher magnification in Figure 6b,d

show clear images ranging from spherical to rod-like and sheet-like structures. These findings were consistent with those of earlier research [47,48].

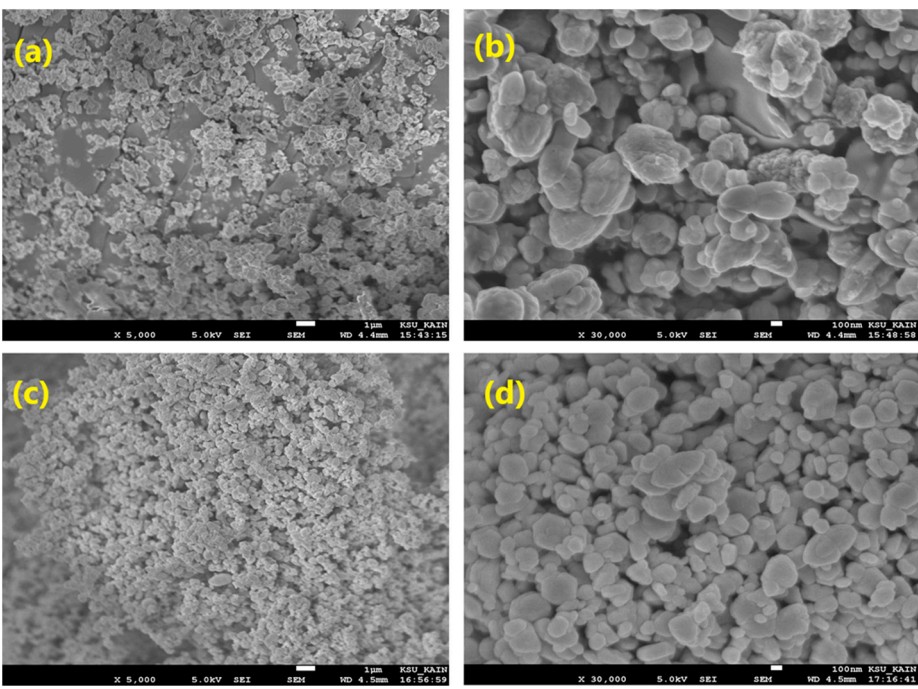

**Figure 6.** SEM images (**a**,**b**) for the without-extract ZnO NPs, and (**c**,**d**) for the green ZnO NPs.

### 3.2.4. EDAX Analysis

X-ray (EDX) techniques were used to further explore the samples, in order to gain additional insight into the topographies of the ZnO NPs. The spectra shown in Figure 7 show three distinct zinc peaks, at the energies of 1 keV, 8.7 keV, and 9.8 keV, respectively, as well as a single oxygen peak at 0.5 keV, all related to ZnO nanoparticles. The majority of the sample was ZnO, as seen by the zinc and oxygen peaks' high intensities [49]. The weight value of 10.6% represents the presence of platinum, which is used in sample coating during SEM imaging, and is often applied to the surface of the sample before SEM examination, as this prevents damage due to the expansion and contraction caused by the electron beam [50].

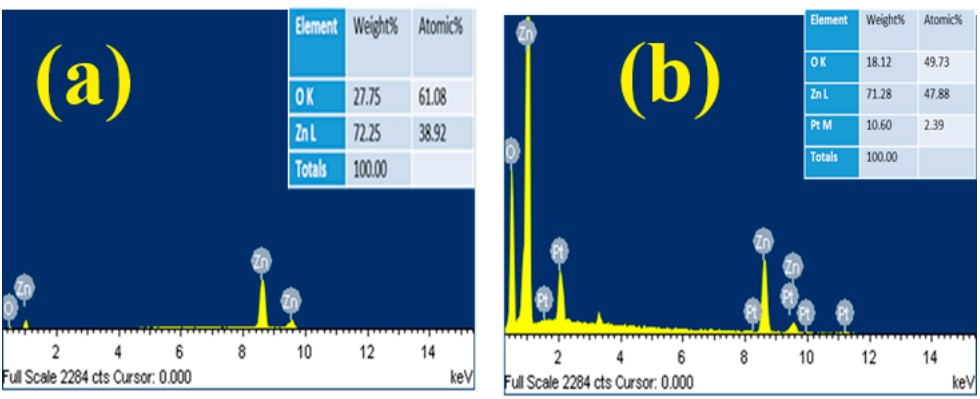

**Figure 7.** EDX spectra of the (**a**) ZnO (without extract) and (**b**) green ZnO NPs.

### 3.2.5. DRS Analysis

The optical characteristics and band-gap energy values of the produced NPs are displayed in Figure 8a,b. The absorption bands for the ZnO (without extract) and the green ZnO NPs were observed at 346 and 366 nm, respectively. These results are consistent

with those reported in the literature [51]. The band-gap energy values of the samples were calculated by expanding the graph's linear component, and plotting $(\alpha h\upsilon)^2$ versus energy (Eg), as shown in Figure 8b. The Tauc equation was used to determine the samples' band-gap energy values [51]:

$$\alpha(h\upsilon)^2 = A(h\upsilon - Eg) \tag{4}$$

Here, $\alpha$ is the absorption coefficient, *h* is Planck's constant, $\upsilon$ is the frequency, Eg is the band-gap energy, and A is a proportionality constant. For the ZnO (without extract) and the green ZnO NPs, the computed Eg values were 2.79 and 2.65, respectively [52]. From the results obtained for the band-gap energy, it is notable that there is a decrease in the Eg values, and this is expected, as some plant extract components overlay or modify the surface, and reduce the band-gap of the nanoparticles. In general, green synthesized nanoparticles are more reactive than their equivalents generated through conventional methods. The overall higher reactivity of particles in the quantum zone is caused by an increase in the electron densities at the lower energy bands, due to the decreased separation of energy states; the study's findings are consistent with those of earlier research [53].

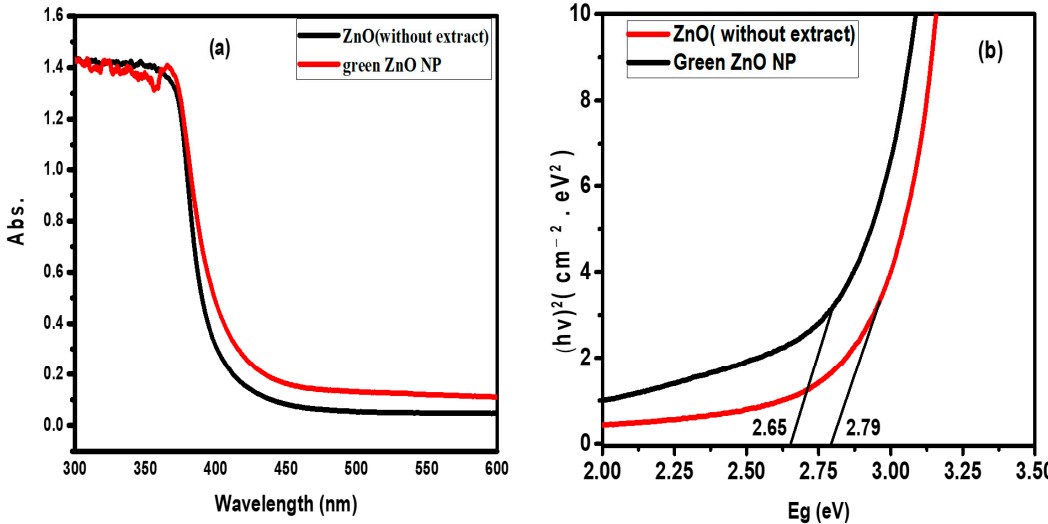

**Figure 8.** (**a**) UV–Vis spectra, and (**b**) Tauc plots of the ZnO (without extract) and green ZnO NPs.

### 3.2.6. BET Determination

The surface area (BET) was used to calculate the precise surface area of the prepared nanomaterial. Figure 9 shows the nitrogen adsorption–desorption isotherms of the produced nanomaterial. The isotherm shows how the NPs behaved as a typical type (IV) isotherm, with a deceleration loop in the low-pressure area ($P/P^0 < 0.8$), which is characteristic of mesoporous nanostructures. Table 5 shows the BET, pore volume, and average pore diameter values. The BET values calculated were 4.003 and 6.032 ($m^2/g$), whereas the pore volumes were 0.011 and 0.017 ($cm^3/g$) for the ZnO (without extract) and the green ZnO NPs, respectively. As the small porous samples had a large surface area, the behavior of the material as a whole may have begun to be dominated by its surface qualities; in comparison, the green ZnO was highly porous, and thus had a high surface area. Figure 9 show the nitrogen adsorption–desorption isotherms of ZnO (without extract) and the green ZnO NPs, whereas Figure 10 illustrates the common Barrett–Joyner–Halenda (BJH) desorption pore size distribution curves for the ZnO (without extract) and the green ZnO NPs. The curves show that the majority of the mesoporous particles had a size of less than 40 nm.

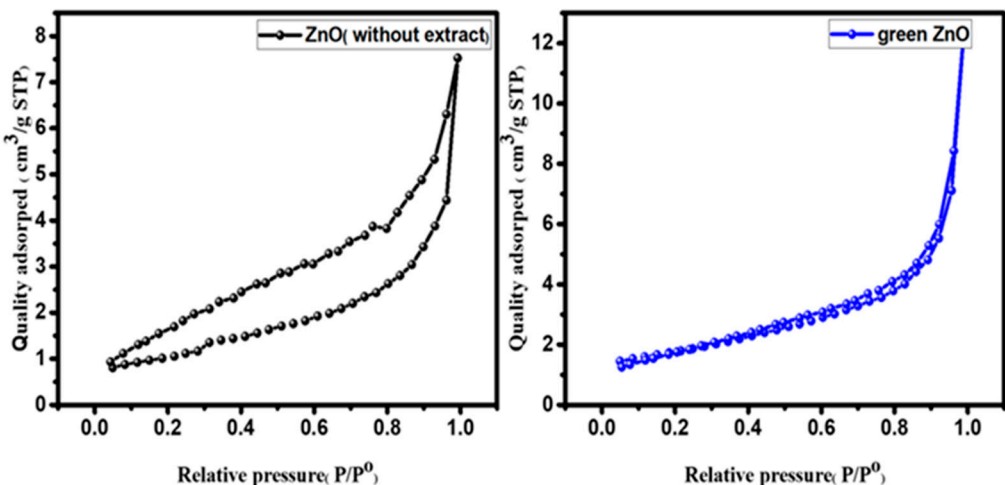

**Figure 9.** $N_2$ adsorption–desorption isotherms of the ZnO (without extract), and the green ZnO NPs.

**Table 5.** BET, pore volume, and pore distribution of the ZnO (without extract) and green ZnO NPs.

| Samples | BET $(m^2/g)$ | Pore Volume $(cm^3/g)$ | Average Pore Diameter (nm) |
|---|---|---|---|
| ZnO (without extract) | 4.003 | 0.011 | 18.455 |
| Green ZnO | 6.032 | 0.017 | 15.876 |

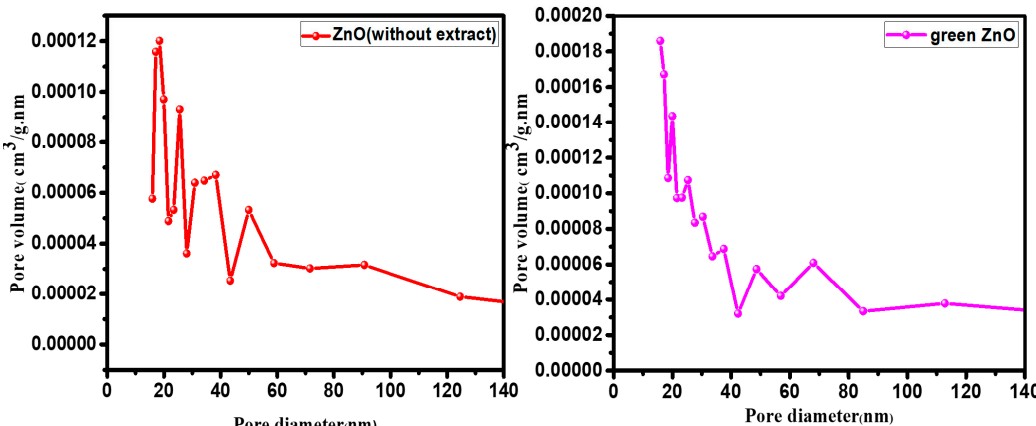

**Figure 10.** BJH pore-size distribution curves for the ZnO (without extract) and green ZnO NPs.

## 4. Conclusions

This study used a green technique to create ZnO NPs, using *Artemisia absinthium* plant extract as a minimizing and stabilizing agent. The characteristics and quality of the green ZnO and the ZnO (without extract) NPs were examined, utilizing several analytical techniques. The study showed that the plant extract from *A. absinthium* efficiently performed a role in decreasing and stabilizing the NP samples that were prepared. The outcomes of this approach yielded spherical and hexagonal forms, with an average size of 18.77 and 24.39 nm for the green ZnO and ZnO (without extract) NPs, respectively. FTIR analysis confirmed the formation of NPs with the new broad absorption bands observed at 432.05 and 439.77 $cm^{-1}$ for the green ZnO and ZnO (without extract) NPs, respectively. The band-gap energy values narrowed from 2.79 to 2.65 between the ZnO (without extract) and green ZnO NPs, respectively.

The $N_2$ adsorption–desorption isotherms demonstrated that the pore volume on the given surface increased with an increasing surface area, from 4.003 to 6.032 $m^2/g$. The

pore volumes were from 0.011 to 0.017 cm$^3$/g, and the pore diameters were from 18.455 to 15.876 nm for the ZnO (without extract) and green ZnO NPs, respectively.

This research demonstrated that using a green approach can generate stable nanosized ZnO particles, while being cost-effective and yielding high-quality crystallization.

This study proved that *A. absinthium* leaf extract is a promising option for green synthesizing ZnO NPs, because it contains essential phytochemicals that can serve as a platform, acting as reducing and capping agents in the process. The uniqueness and importance of this study lie in the green synthesis of ZnO nanoparticles from *A. absinthium* plant leaf extract, which have potential applications in the pharmaceutical (antibacterial) industry, cosmetic, anticancer, antidiabetic, opto-electronic (solar cell), concrete, and rubber industries, as well as in the treatment of water pollution (dye removal), photocatalysis, and photodegradation.

**Author Contributions:** Conceptualization F.N.A. and Z.M.A.; methodology, F.N.A.; software, Z.M.A.; formal analysis, F.N.A.; investigation, Z.M.A.; resources, F.N.A.; data curation S.Z.A.M.; writing—original draft preparation, F.N.A.; writing—review and editing, Z.M.A. and S.Z.A.M.; supervision, Z.M.A. and S.Z.A.M.; project administration, Z.M.A.; funding acquisition, F.N.A. All authors have read and agreed to the published version of the manuscript.

**Funding:** This research received no external funding.

**Institutional Review Board Statement:** Not applicable.

**Informed Consent Statement:** Not applicable.

**Data Availability Statement:** All data are available on request.

**Acknowledgments:** The authors are grateful for the support of the Chemistry Department in the College of Science and Arts at Al-Rass, Qassim University.

**Conflicts of Interest:** All authors confirm that they do not have any conflicts of interest related to the research in this manuscript.

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
