# Peer review of "Phytochemical Substances—Mediated Synthesis of Zinc Oxide Nanoparticles (ZnO NPS)"

_inorganics, doi:10.3390/inorganics11080328_

Round 1

Reviewer 1 Report

This paper introduces interesting research on the synthesis of Zinc Oxide Nanoparticles (ZnO NPS). Zinc oxide nanoparticles was successfully fabricated by using the Artemisia absinthium (A. absinthium) leaf extract. The paper can be accepted after the following minor modifications.

(1) The data in the article should be presented in a more standard way and more attention should be paid to retaining Significant Digit.

(2) Table 1 should be presented using a three-line table.

(3) The words and numbers in Figure 9 and Figure 10 are too small and not clear enough.

Author Response

Dear reviewer;

Thank you for reviewing the manuscript, I upload the response with the manuscript.

Reviewer 2 Report

Review for the manuscript:

Entitled: "Phytochemical Substances —Mediated Synthesis of Zinc Oxide Nanoparticles (ZnO NPS) "

for Inorganics.

With ID: inorganics-2511951

General comments

Comments for the Authors

This work is well within the scope of Inorganics, and it may be of interest to most of the readers of this journal. It is very well organized with good references to follow. Turnitin shows a similarity index of 27% across text, even in the results section. Some examples of duplicated text come from the articles below:

DOI: https://doi.org/10.47352/jmans.v1i1.9

DOI: https://doi.org/10.1007/s41204-019-0057-3

https://www.biorxiv.org/content/10.1101/2022.10.27.514023v1.full

http://doi.org/10.17576/jsm-2022-5102-17

https://doi.org/10.1016/j.plaphy.2016.06.001

Turnitin showed that 0% of qualifying text in this submission has been determined to be generated by AI.

Thus, authors should reduce the overlapping text and provide better justification regarding the novelty and the importance of this work compared to already published ones, from their group and others.

For all the above, and the specific comments below, I have opted to recommend a Major revision for the current form of this work.

Specific comments

Introduction

P1: ‘metal NPs’ Please define every abbreviation the first time cited in the text.

Materials and Methods

The Equations in P4 and P6 do not appear correctly. Furthermore, what is the purpose to show this equation in P6 again since it is the same as in P4? Please revise. Furthermore, please provide a reference for the statement: ‘K is a constant equal to 0.90’.

Results

P8, Fig7. There is a 10.6% weight in the second fig. Please explain.

Author Response

(The authors gave the same response as above.)

Reviewer 3 Report

Report on the manuscript inorganics-2511951 entitled “Phytochemical Substances —Mediated Synthesis of Zinc Oxide Nanoparticles (ZnO NPS)”.

The submitted manuscript should be revised. In this work, ZnO NPs were prepared from Artemisia absinthium leaf extract and characterized via XRD, SEM, EDX, FT-IR, and optical spectroscopy. The thought of paper is old and was published before with application [https://doi.org/10.1016/j.envpol.2023.121105]. In addition, the following points should be addressed:

1. The language of the manuscript should be revised.

2. In introduction “Many engineering applications exist for ZnO NPs, including solar cells [7], photodetectors, biosensors [8], chemical sensors [9], and gas sensors.”, the major application of ZnO is photodegradation or water treatment so, it should be added and the recommended references to confirm this “[Colloids and Surfaces A: Physicochemical and Engineering Aspects, Volume 618, 5 June 2021, 126437] and [Journal of Alloys and Compounds, Volume 886, 15 December 2021, 161169] & …etc”

3. The XRD analysis should have the expected JCPDS card number and the calculations of crystal size hasn’t any discussion which should indicates the reason of increasing crystal size. The a, b, and c parameters should be calculated to clearly compare between ZnO NPs (without extract) and green ZnO NPs

4. There are many organic peaks in FTIR of ZnO and this should be explained why and how it formed?

5. The SEM images in figure 6 has low-resolution. Please, enhance it?

6. The EDX analysis shows the different in chemistry between ZnO NPs (without extract) and green ZnO NPs as the O-percent is high in case of ZnO without extract and O-percent is low in the case of using extract. This result is in conflict with what obtained from FTIR that having the same bonds!

7. For the ZnO (without extract) and green ZnO NPs, the computed Eg values were 2.79 and 2.65, respectively. The authours should indicate the reason for this.

8. What is the application of the prepared ZnO?

9. Conclusion should be clearer and it should have clearly what achieved in this work.

 Extensive editing of English language requiredز

Author Response

Dear reviewer

Thank you for reviewing the manuscript, I upload the response with the revised manuscript.

Round 2

Reviewer 2 Report

Review for the manuscript:

Entitled: "Phytochemical Substances —Mediated Synthesis of Zinc Oxide Nanoparticles (ZnO NPS) "

for Inorganics.

With ID: inorganics-2511951.R1

General comments

Comments for the Authors

Authors responded to my previous remarks thus the manuscript can be published.

Best regards

Reviewer 3 Report

It could be accepted.